# AmoebaLLM: Constructing Any-Shape Large Language Models for Efficient and Instant Deployment

**Yonggan Fu, Zhongzhi Yu**∗**, Junwei Li**∗**, Jiayi Qian**∗**, Yongan Zhang, Xiangchi Yuan,**
**Dachuan Shi, Roman Yakunin, Yingyan (Celine) Lin**
Georgia Institute of Technology
{yonggan.fu, celine.lin}@gatech.edu

## Abstract

Motivated by the transformative capabilities of large language models (LLMs) across various natural language tasks, there has been a growing demand to deploy these models effectively across diverse real-world applications and platforms. However, the challenge of efficiently deploying LLMs has become increasingly pronounced due to the varying application-specific performance requirements and the rapid evolution of computational platforms, which feature diverse resource constraints and deployment flows. These varying requirements necessitate LLMs that can adapt their structures (depth and width) for optimal efficiency across different platforms and application specifications. To address this critical gap, we propose AmoebaLLM, a novel framework designed to enable the instant derivation of LLM subnets of arbitrary shapes, which achieve the accuracy-efficiency frontier and can be extracted immediately after a one-time fine-tuning. In this way, AmoebaLLM significantly facilitates rapid deployment tailored to various platforms and applications. Specifically, AmoebaLLM integrates three innovative components: (1) a knowledge-preserving subnet selection strategy that features a dynamic-programming approach for depth shrinking and an importance-driven method for width shrinking; (2) a shape-aware mixture of LoRAs to mitigate gradient conflicts among subnets during fine-tuning; and (3) an in-place distillation scheme with loss-magnitude balancing as the fine-tuning objective. Extensive experiments validate that AmoebaLLM not only sets new standards in LLM adaptability but also successfully delivers subnets that achieve state-of-the-art trade-offs between accuracy and efficiency. Our code is available at https://github.com/GATECH-EIC/AmoebaLLM.

## 1 Introduction

The remarkable abilities and transformative impacts of large language models (LLMs) [1, 2, 3, 4] have been paralleled by a growing interest in deploying them across a wide range of real-world applications and diverse platforms. However, given the rapid evolution of computational platforms and varying application-specific requirements, the challenge of deploying LLMs efficiently on various platforms with differing specifications has become more pronounced. This is because diverse platforms often feature different resource constraints and deployment flows, necessitating LLMs with varying structures and shapes (i.e., depth and width) to achieve maximized execution efficiency, as affirmed by our profiling in Sec. 2. Moreover, even the same platform may have varying requirements for LLMs' execution efficiency depending on factors such as on-device battery status. These varying requirements demand a flexible framework capable of adapting to both the intrinsic hardware constraints and the extrinsic demands of diverse application scenarios.

---

∗Contributed equally.

38th Conference on Neural Information Processing Systems (NeurIPS 2024).

Existing efficient LLM solutions [5, 6, 7, 8, 9, 10], which primarily use model compression to bridge the gap between the resource constraints of the target device and the prohibitive complexity of LLMs, fail to fully address the above needs. This is because these solutions either focus on a single dimension of compression, resulting in limited efficiency improvements, or require a costly fine-tuning process for each target platform with its unique specifications. This strategy is particularly unscalable and inefficient for deploying widely used public LLMs like LLaMA [1, 2], where each user must compress and fine-tune the LLMs for their specific platform and application needs.

In light of this, it is highly desirable to develop a suite of LLMs designed such that compressed subnets of arbitrary shapes, which can achieve the accuracy-efficiency frontier without the necessity of individual fine-tuning, can be instantly extracted, thus allowing for immediate adaptation to the diverse needs of various platforms and applications. To achieve this, previous one-for-all training techniques [11, 12, 13, 14, 15], which strategically sample subnets for joint training to deliver models with switchable complexity, are promising candidates. However, directly applying them to pre-trained LLMs would lead to failure due to the following challenges: (1) their adopted subnet sampling strategies, which are dedicated to models trained from scratch, are not applicable to extensively pre-trained LLMs as informative and critical components that store useful knowledge are highly likely to be skipped; (2) jointly fine-tuning different subnets on commonly adopted small-scale tuning datasets can easily cause severe gradient conflict [16, 17] for LLM weights pre-trained on a large corpus, thus leading to poor performance of all subnets.

To address these challenges, we propose a framework called AmoebaLLM, which endows a given LLM with the ability to instantly derive compressed subnets of arbitrary shapes that can achieve the accuracy-efficiency frontier. We achieve this through the development of three key components of AmoebaLLM's one-for-all fine-tuning scheme: a subnet selection strategy, a trainable adapter design, and a fine-tuning objective. Specifically, we summarize our contributions as follows:

- We develop a framework called AmoebaLLM, which grants a given LLM the capability to deliver subnets of arbitrary shapes that achieve state-of-the-art (SOTA) accuracy-efficiency trade-offs after a one-time fine-tuning. In this way, AmoebaLLM can greatly facilitate rapid deployment across varying platforms and applications. This is achieved by integrating the following three key components to enable one-for-all fine-tuning.

- For extracting high-quality subnets with diverse shapes, we propose a knowledge-preserving subnet selection strategy that features dynamic programming (DP)-based depth shrinking and importance-driven width shrinking. This addresses the aforementioned challenge (1) by preserving the encoded factual knowledge and reasoning capabilities of pre-trained LLMs.

- For the trainable adapter during one-for-all fine-tuning, we propose a shape-aware mixture of LoRAs (SMoL), which selects and combines a sparse set of LoRAs [18] using a gating function that takes the subnet shape as input. This technique addresses the aforementioned challenge (2) by mitigating gradient conflicts among subnets. More importantly, once the target subnet shape is determined based on the target platform/application at deployment time, all the selected LoRAs can be merged into the LLM weights, thus eliminating overhead.

- For the fine-tuning objective, we enhance the in-place distillation strategy [11, 13, 14] by integrating a loss-magnitude balancing scheme. This scheme is based on the observation that the loss magnitudes of subnets with different shapes in pre-trained LLMs are unbalanced, leading to a bias toward specific subnets and thus poor overall performance. Our proposed technique effectively addresses this issue and improves the performance of all subnets.

- Extensive experiments validate that our AmoebaLLM can deliver LLMs with instantly serviceable subnets of any shape, each performing better or on par with SOTA efficient LLM solutions. Additionally, when leveraging AmoebaLLM as a pure LLM compression framework to compress a given LLM to the target parameter, it can achieve new SOTA compression effectiveness, thanks to its subnet selection strategy.

## 2    Motivation and Profiling

Before introducing our framework, we first conduct a profiling of generation latency across different devices, deployment flows, and use cases to examine the demand for LLMs with adaptable structures that meet the diverse needs of various platforms and applications.

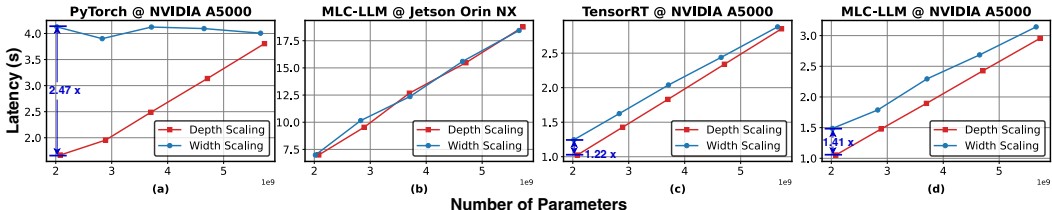

Figure 1: The latency of LLaMA2 7B with scaled depth/width on various devices/deployment flows.

**Profiling setup.** We study the efficiency of different LLM shapes on different devices by uniformly scaling either the depth or width of LLaMA2 7B [1] to the same model size. Here, depth is defined as the number of self-attention blocks, each including both a multi-head attention module and a feed-forward network, while width is defined as the hidden dimensions. We profile these workloads using (1) two devices, including an NVIDIA A5000 consumer-level GPU and an NVIDIA Jetson Orin NX edge GPU; and (2) three deployment flows, including TensorRT-LLM [19], MLC-LLM [20], and vanilla PyTorch [21].

**Observation and analysis.** As shown in Fig. 1, we can observe that (1) first of all, the same workload on two GPUs with different resources exhibits large latency gaps, indicating the need for LLMs with adaptable structures to adapt to different devices when aiming to ensure a comparable latency to satisfy user needs; (2) the preference regarding LLM shapes differs across deployment flows. Specifically, under the same model size, reducing model depth and width have a comparable impact on the measured latency using TensorRT-LLM on A5000 and MLC-LLM on Orin NX, while reducing depth using PyTorch or MLC-LLM on A5000 can achieve a notably lower latency than reducing width. This implies that for emerging platforms with limited compatible deployment flows, proper customization of LLM shapes for maximized efficiency is needed.

## 3 The Proposed AmoebaLLM Framework

### 3.1 AmoebaLLM: Methodology Overview

To address the challenges associated with traditional one-for-all network training, as mentioned in Sec. 1, our AmoebaLLM is equipped with three components: a knowledge-preserving subnet selection strategy, an SMoL adapter, and an in-place distillation fine-tuning objective with loss-magnitude balancing. We illustrate our overall framework in Fig. 2: given a target LLM, our AmoebaLLM endows it with the capability of instantly deriving capable subnets via a two-stage process.

In the first stage, AmoebaLLM generates the subnet selection strategy. Specifically, given the target depth/width remaining ratios, this step decides which layers/neurons to maintain, respectively. To maximally preserve the knowledge and language modeling capabilities of pre-trained LLMs, we propose employing dynamic programming [22] to determine the retained layers under different remaining ratios and leverage neuron importance metrics [8] to retain important neurons in a structured manner, as detailed in Sec. 3.2. After this stage, the subnet selection strategy is determined and fixed.

In the second stage, we insert our proposed SMoL adapter into the target LLM for a one-time, one-for-all fine-tuning. Specifically, SMoL is composed of a set of LoRAs [18] and employs a gating function to sparsely activate different subsets of LoRAs for different subnets, thus mitigating their gradient conflicts. This process is elaborated upon in Sec. 3.3. In addition, our fine-tuning objective enhances the sandwich sampling and in-place distillation methods described in [11, 13, 14] by adding a loss-magnitude balancing scheme. This prevents bias towards specific subnets in pre-trained LLMs, as detailed in Sec. 3.4. At deployment time, the selected LoRAs, which are determined by the extracted subnet shape favorable to the target platform, can be merged into the LLM weights.

### 3.2 AmoebaLLM: The Proposed Knowledge-Preserving Subnet Selection Strategy

**Motivation.** As detailed in Sec. 5, previous one-for-all training techniques [11, 12, 13, 14, 15] often select the first layers of a model or the first channels of a layer. These techniques, intended for models trained from scratch, are unsuitable for pre-trained LLMs with rich knowledge encoded in their weights. Considering the difficulty of recovering lost knowledge through fine-tuning or our one-for-all fine-tuning on a relatively small corpus [23], it is crucial to identify informative and critical

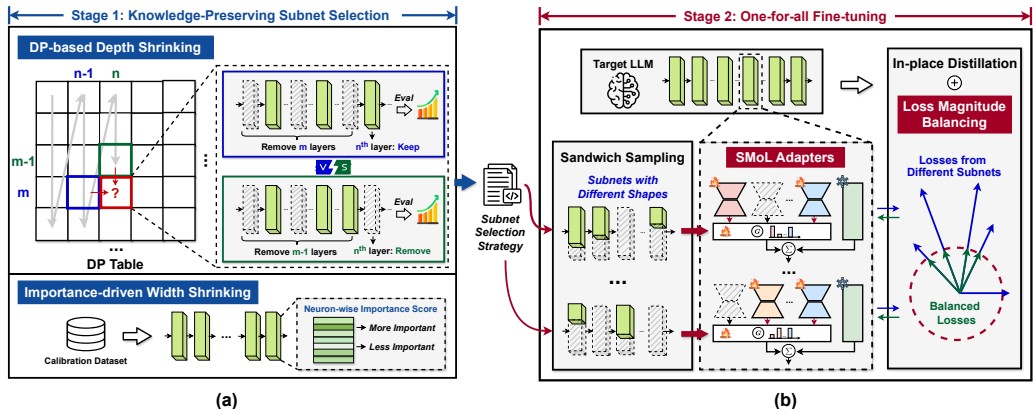

Figure 2: An overview of our AmoebaLLM framework: (a) Stage 1: Generate the subnet selection strategy; (b) Stage 2: One-for-all fine-tuning. Zoom in for a better view.

layers/neurons that store useful knowledge during the subnet selection process instead of relying solely on fine-tuning. To this end, we propose our knowledge-preserving subnet selection strategy to select the most informative layers/neurons under a given remaining ratio, as detailed below.

**DP-based depth shrinking.** Previous works have made diverse observations regarding the layer locations that store knowledge in different series of language models [24, 25, 26, 27, 28, 29]. As such, it is highly desirable to have a principled strategy to evaluate the joint contributions of different layer combinations in a target LLM to derive the optimal layer selection strategy for each remaining ratio. To achieve this goal, we propose a DP-based depth shrinking strategy.

*Problem formulation.* Given a target LLM with $N$ decoder layers, we define the layer selection strategy by a vector $s \in \{0,1\}^N$. Here, $s[n] = 1$ indicates that the $n$-th layer is retained; otherwise, the layer is removed. The objective is to determine the selection strategy $s$ that achieves the optimal target metric, such as maximal accuracy or minimal perplexity (PPL), on a calibration dataset $C$, subject to the constraint that $M$ layers are removed.

*Key hypothesis.* Thanks to the residual structure [30] of common LLMs [1, 2, 3] and the observations that LLMs' knowledge is compositional across layers [24, 31, 32], we hypothesize that the layer selection problem described above can be divided into smaller and *approximately* independent sub-problems. Consequently, we can employ dynamic programming [22] to effectively and efficiently solve the layer selection problem.

*Our DP-based methodology.* We define a DP table $\mathbf{D}[n][m]$ (where $n \in [1, N]$ and $m \in [1, M]$), which stores the best target metric on the calibration dataset when exactly $m$ layers are removed from the **first** $n$ layers of the target LLM. The corresponding layer selection strategy is denoted as $\mathbf{S}[n][m] \in \{0,1\}^N$. Consequently, $\mathbf{S}[N][M]$ represents the final strategy derived for removing $M$ layers out of all $N$ layers. Next, we elaborate on how to obtain $\mathbf{D}[n][m]$ and $\mathbf{S}[n][m]$, where we assume the target metric is such that larger values are better, without losing generality.

As illustrated in Fig. 2 (a), similar to general DP problems [22], $\mathbf{D}[n][m]$ can be derived by a recurrence relationship. Specifically, to derive each $\mathbf{D}[n][m]$, we compare the metrics achieved by the following two cases: (1) removing $m$ layers from the first $n-1$ layers, and (2) removing $m-1$ layers from the first $n-1$ layers and removing the $n$-th layer. The strategy yielding better metrics is adopted. More formally, this process can be formulated as follows:

$$\mathbf{D}[n][m] = \max\left(\mathbf{D}[n-1][m], \mathbf{P}(n, m)\right) \tag{1}$$

where $\mathbf{P}(n, m)$ is the metric obtained by removing the $n$-th layer on top of the best-known strategy $\mathbf{S}[n-1][m-1]$ for removing $m-1$ layers from the first $n-1$ layers. This is computed as follows:

$$\mathbf{P}(n, m) = evaluate(remove(\mathbf{S}[n-1][m-1], n), C) \tag{2}$$

where $remove(s, n)$ is a function that sets the $n$-th layer to 0 in a strategy $s$. Leveraging this recurrence relationship, after initializing the DP table with the base cases, i.e., $\mathbf{D}[i][0] = \infty$ and $\mathbf{S}[i][0] = \{1\}^N$ ($\forall i \in [1, N]$), since no layer is removed, the full DP table can be established with a complexity of $\mathbb{O}(MN)$. In practice, we set $M$ as the maximum number of layers allowed to be removed in our

design space. Therefore, after obtaining the corresponding DP table $\mathbf{D}$, the layer configuration of any subnet with $m$ layers removed ($\forall m \in [1, M]$) can be directly obtained from $\mathbf{D}[N][m]$. Note that constructing the DP table is a one-time effort for a given LLM.

*Differences from and connections with previous methods.* The most relevant direction is layer pruning for LLMs. Pioneering works along this direction either utilize single-layer importance to determine which layers to prune [9], failing to measure the layers' joint contribution and thus aggressively losing factual knowledge as shown in Sec. 4, or rely on pre-defined rules [33], i.e., removing consecutive layers among the last ones, which is suboptimal and may not be generalizable to future LLMs. In contrast, our DP-based strategy can measure the joint impact of different layer combinations and is principled and fully automated without relying on human priors for specific LLMs. *More importantly*, the two aforementioned works [9, 33] are subsets of our DP-based strategy's solution space. Thus, it can serve as a new SOTA layer pruning method, as demonstrated in Sec. 4.3, in addition to serving as a component (i.e., the subnet selection strategy) in one-for-all fine-tuning. Previous works [34, 35] have also applied dynamic programming for pruning at different granularities with varied formulations, and our work is the first to introduce this classical approach to LLMs.

**Importance-driven width shrinking.** Compared to layer selection, removing neurons (and all corresponding weight connections) for width shrinking is more fine-grained and involves a much larger design space. Therefore, instead of using dynamic programming, we directly employ existing structured pruning metrics for LLMs to obtain importance scores for each neuron and select the most important ones during one-for-all training, given a width remaining ratio.

To achieve this, we employ the metric in [8] due to its outstanding performance compared to other alternatives. Specifically, we adopt the same width remaining ratio for linear layers within the same block (either the self-attention block or the feed-forward network), where the importance scores of input neurons of the last linear layer in each block are used to determine the width configuration of this block. In other words, when a subset of input neurons to the last linear layer is removed, all associated neurons and weights [7] within this block will be removed in a structured manner.

*Importance metric.* The importance score $\mathbf{F}_i^\ell$ of the $i$-th input neuron in the last linear layer $\mathbf{W}^\ell$ of the $\ell$-th block is computed as $\mathbf{F}_i^\ell = \underset{k,t,j}{\mathbb{E}}(\mathbf{X}_{k,t,i}^\ell - \overline{\mathbf{X}}_i^\ell)^2 \cdot \|\mathbf{W}_{j,i}^\ell\|_2^2$, where $j$ is the index of the output neurons, $\mathbf{X}_{k,t,i}^\ell$ is the input features of the $t$-th token in the $k$-th batch, and $\overline{\mathbf{X}}_i^\ell$ is the averaged input features across these two dimensions, both received by the $i$-th input neuron. To better maintain the LLMs' capability under a given remaining ratio: (1) the importance score is further normalized over all input neurons in each layer and globally sorted for non-uniform width shrinking; (2) a pre-computed bias term $\mathbf{B}^\ell$ is added to the output neurons to compensate for the removed input neurons, i.e., $\mathbf{B}^\ell = \mathbf{W}^\ell((1 - \mathbf{M}^\ell) \odot \overline{\mathbf{X}}^\ell)$, where $\mathbf{M}^\ell$ is the binary mask indicating whether the input neurons are retained. We refer the readers to [8] for more details.

## 3.3 AmoebaLLM: The Proposed Shape-aware Mixture of LoRAs

**Motivation.** As demonstrated in Sec. 4.3, joint weight fine-tuning of different subnets on small-scale datasets can lead to severe gradient conflicts [16, 17], resulting in poor performance across all subnets. One potential solution is to adopt parameter-efficient adapters like LoRA [18]. However, accumulating gradients from all subnets onto the same LoRA still suffers from gradient conflicts, making fine-tuning unstable and causing some subnets to underperform. On the other hand, tuning a separate LoRA for each subnet configuration is infeasible due to the large design space. To address this, we propose an intermediate solution to balance performance and efficiency: the SMoL adapter that features a set of LoRAs, which are sparsely activated based on the subnet shape.

**SMoL adapter design.** Our SMoL adapter consists of a set of $T$ LoRAs $\{\mathbf{\Delta W}_i = \mathbf{B}_i \mathbf{A}_i\}_{i=1}^T$, which are sparsely activated for each subnet shape using a gating function $\mathbf{G}$. Specifically, we employ a one-hot mask $\mathbf{M}$ to indicate the shape, i.e., the layer/width configuration, of the subnet. This mask is fed into the gating function $\mathbf{G}$ to calculate a score for each LoRA. Only the top $k$ LoRAs are activated and weightedly averaged for each subnet shape during fine-tuning, thus mitigating the gradient conflicts among different subnets.

**Implementation.** We extend the noisy top-K gating mechanism from [36] to implement our SMoL. Specifically, $\mathbf{G}(\mathbf{M}) = \text{Softmax}(\text{KeepTopK}(\mathbf{H}(\mathbf{M}), k))$, where $\text{KeepTopK}(\mathbf{v}, k)_i$ is $v_i$ if $v_i$ is in the

top $k$ of $\mathbf{v}$ and $-\infty$ otherwise, and $\mathbf{H}(\mathbf{x})_i = (\mathbf{x} \cdot \mathbf{W}_g)_i + std \cdot \text{Softplus}((\mathbf{x} \cdot \mathbf{W}_{\text{noise}})_i)$. Here, $\mathbf{W}_g$ and $\mathbf{W}_{\text{noise}}$ are learnable, with the latter used to control the noise *std* for load balancing [36]. The final composited weight $\mathbf{W} = \mathbf{W}_{base} + \sum_{i=1}^{T} \mathbf{G}(\mathbf{M})_i \mathbf{\Delta W_i}$, where $\mathbf{W}_{base}$ is the pre-trained weight.

**Key difference from previous works.** Previous mixture-of-LoRA designs [37, 38, 39] are input-adaptive at inference time, and thus the weights of different LoRAs cannot be merged into the original model. In contrast, our SMoL depends only on the subnet shape and is independent of model inputs. This implies that once the target subnet is extracted based on the target platform, all activated LoRAs can be merged into the model weights, thus enhancing parameter efficiency at deployment time.

### 3.4 AmoebaLLM: The Fine-tuning Objective

**Motivation.** Strategically sampling and jointly fine-tuning different subnets is necessary to ensure high-performance subnets within the same LLM, where the sandwich sampling and in-place distillation mechanisms [11, 13, 14] can serve as promising sampling and fine-tuning schemes, respectively. However, naively doing so can cause the larger subnets to gradually underperform during fine-tuning. We identify that this is due to notable differences in the loss magnitudes of different subnets, with smaller subnets, which diverge more from the well pre-trained full model, exhibiting much higher losses than larger subnets. To address this, we propose a simple but effective solution: equipping the in-place distillation with a loss-magnitude balancing mechanism, which we elaborate on as follows.

**In-place distillation with loss-magnitude balancing.** During each fine-tuning iteration, we employ the sandwich sampling [11, 13, 14] to sample $K$ subnets $\{\mathcal{T}_i\}_{i=1}^{K}$ with different layer/width remaining ratios, including the largest/smallest ones and $K-2$ random ones from our design space. Detailed layer/width configurations of sampled subsets can be obtained from the strategies derived in Sec. 3.2. We fine-tune our SMoL adapter as detailed in Sec. 3.3 by accumulating the gradients from all sampled subnets using in-place distillation, where only the loss of the largest subnet $\mathcal{T}_1$ is calculated using ground truth, while those of other subnets $\{\mathcal{T}_i\}_{i=2}^{K}$ use distillation from the largest one [11]. To balance the loss magnitude from different subnets, we normalize all subnets' loss magnitudes to that of the largest subnet, as visualized on the rightmost side of Fig. 2 (b). In this way, the final loss direction is jointly determined by all subnets' loss directions, falling on a unisphere without being severely impacted by their unbalanced magnitudes. We formulate the fine-tuning objective as follows:

$$\mathcal{L}_{\text{total}} = \mathcal{L}_{\text{CE}}(\mathcal{T}_1(x), y) + \sum_{i=2}^{K} \frac{\|\mathcal{L}_{\text{CE}}(\mathcal{T}_1(x), y)\|}{\|\mathcal{L}_{\text{CE}}(\mathcal{T}_i(x), \mathcal{T}_1(x))\|} \mathcal{L}_{\text{CE}}(\mathcal{T}_i(x), \mathcal{T}_1(x)) \tag{3}$$

where $x$ and $y$ are the input and ground truth, respectively, and $\mathcal{L}_{\text{CE}}$ is a cross-entropy loss function.

**Final subnet search after fine-tuning**. We adopt a simple hierarchical search strategy to select subnets from the fine-tuned LLM to satisfy the target efficiency constraint while maximizing achievable accuracy. Specifically, we first perform a coarse grid search across uniformly spaced depth and width settings based on a small calibration set (e.g., 40 samples from the MMLU dataset) to identify subnets that meet the efficiency constraint with maximized accuracy. Next, we conduct a more fine-grained grid search within depth and width ranges surrounding the optimal subnet identified in the coarse grid search stage. More advanced search strategies, such as evolutionary search [11, 12, 13, 14], are left for future work.

## 4 Experimental Results

### 4.1 Experiment Setup

**Baselines.** Our baselines include two SOTA structured width pruning methods: LLM-Pruner [7] and FLAP [8], and one layer pruning method Shortened LLaMA [9]. All these baselines are open-sourced and we apply their official code to different LLMs. All baselines, including FLAP which were not fine-tuned in their original paper, are fine-tuned using the settings below for a fair comparison.

**Fine-tuning setting.** Following [7, 9], we adopt 50K samples from Alpaca [40] for our one-for-all fine-tuning as well as for fine-tuning all baselines. For both our method and the baselines, we adopt a constant learning rate of 2e-4 with an AdamW optimizer and a LoRA rank of 64, and fine-tune for 10K iterations. It takes 40 GPU hours on an NVIDIA A5000 GPU for our one-for-all fine-tuning.

Table 1: Compare with baseline methods under varying remaining ratios on LLaMA2 7B.

| Ratio | Method ↓ | MMLU ↑ | Average ↑ | BoolQ ↑ | PIQA ↑ | HellaSwag ↑ | WinoGrande ↑ | ARC-e ↑ | ARC-c ↑ | OBQA ↑ |
|---|---|---|---|---|---|---|---|---|---|---|
| 80% | LLM-Pruner [7] | 29.63 | 56.95 | 58.53 | 76.39 | 65.80 | 60.38 | 64.60 | 34.56 | 38.40 |
| | FLAP [8] | 40.21 | 60.98 | **73.64** | 74.81 | 68.27 | 65.43 | 66.20 | 37.88 | 40.60 |
| | Shortened LLaMA [9] | 26.45 | 58.72 | 62.17 | 76.01 | 68.22 | 58.88 | 68.98 | 38.40 | 38.40 |
| | AmoebaLLM (Ours) | 40.70 | 62.29 | 72.70 | **76.80** | 70.60 | **67.60** | 68.30 | 38.80 | **41.20** |
| | AmoebaLLM† (Ours) | 42.40 | **62.37** | 72.50 | 76.30 | **70.80** | 66.90 | **70.30** | **40.20** | 39.60 |
| 65% | LLM-Pruner [7] | 23.15 | 54.09 | 60.73 | **74.97** | 58.57 | 57.85 | 55.68 | 33.02 | 37.80 |
| | FLAP [8] | 33.28 | 56.12 | 65.75 | 70.08 | 60.57 | 61.33 | 62.25 | 33.87 | **39.00** |
| | Shortened LLaMA [9] | 24.89 | 52.57 | 62.32 | 72.03 | 55.10 | 52.41 | 59.47 | 30.63 | 36.00 |
| | AmoebaLLM (Ours) | 36.00 | 56.96 | **72.10** | 70.70 | 59.70 | **63.20** | 62.20 | 34.00 | 36.80 |
| | AmoebaLLM† (Ours) | 36.20 | **57.26** | 70.50 | 70.90 | **61.50** | 62.70 | **63.50** | **34.50** | 37.20 |
| 50% | LLM-Pruner [7] | 22.90 | 47.52 | 61.83 | **67.79** | 43.31 | 51.22 | 46.13 | 28.16 | 34.20 |
| | FLAP [8] | 27.67 | 51.12 | 59.45 | 67.30 | 51.33 | 56.75 | 55.43 | **31.57** | **36.00** |
| | Shortened LLaMA [9] | 24.76 | 47.35 | 62.23 | 66.00 | 43.60 | 51.54 | 50.63 | 26.45 | 31.00 |
| | AmoebaLLM (Ours) | 30.60 | 52.19 | **65.70** | 66.10 | 51.30 | 60.10 | 56.60 | 31.50 | 34.00 |
| | AmoebaLLM† (Ours) | **32.20** | **52.63** | 64.70 | 66.70 | **53.00** | 60.30 | **58.00** | 30.10 | 35.60 |

Table 2: Compare with baseline methods under varying remaining ratios on Vicuna 7B v1.5.

| Ratio | Method ↓ | MMLU ↑ | Average ↑ | BoolQ ↑ | PIQA ↑ | HellaSwag ↑ | WinoGrande ↑ | ARC-e ↑ | ARC-c ↑ | OBQA ↑ |
|---|---|---|---|---|---|---|---|---|---|---|
| 80% | LLM-Pruner | 38.94 | 57.80 | 64.27 | 75.35 | 64.28 | 61.88 | 64.35 | 35.07 | 39.40 |
| | FLAP | 43.50 | 60.54 | 72.45 | 73.67 | 66.59 | 65.98 | 68.14 | 37.54 | 39.40 |
| | Shortened LLaMA | 35.27 | 60.93 | 67.58 | **75.68** | 68.12 | 64.48 | **70.20** | 40.44 | 40.00 |
| | AmoebaLLM (Ours) | 47.40 | 60.77 | 71.30 | 72.70 | **68.80** | **66.10** | 66.50 | 38.60 | **41.40** |
| | AmoebaLLM† (Ours) | 48.30 | **61.54** | **73.10** | 73.33 | 68.30 | 65.80 | 69.30 | **40.80** | 40.20 |
| 65% | LLM-Pruner | 24.07 | 54.24 | 61.19 | **73.50** | 57.67 | 57.85 | 58.67 | 32.42 | **38.40** |
| | FLAP | 38.08 | 56.47 | 68.96 | 70.78 | 58.97 | 61.33 | **63.43** | 34.21 | 37.60 |
| | Shortened LLaMA | 25.59 | 52.50 | 64.28 | 70.62 | 56.78 | 57.46 | 52.74 | 31.40 | 34.20 |
| | AmoebaLLM (Ours) | 40.30 | 55.93 | 68.20 | 69.80 | 58.60 | 62.40 | 62.20 | 34.10 | 36.20 |
| | AmoebaLLM† (Ours) | 44.60 | **56.74** | **71.50** | 69.60 | **60.30** | **64.50** | 61.30 | **34.40** | 35.60 |
| 50% | LLM-Pruner | 23.24 | 47.98 | 59.08 | **68.55** | 44.24 | 52.17 | 49.03 | 28.41 | 34.40 |
| | FLAP | 29.92 | 50.74 | 56.15 | 67.52 | **51.81** | 57.69 | 56.57 | **31.06** | 34.40 |
| | Shortened LLaMA | 25.03 | 45.56 | 51.71 | 65.67 | 43.28 | 51.38 | 49.96 | 26.54 | 30.40 |
| | AmoebaLLM (Ours) | 34.00 | 51.41 | 64.20 | 65.00 | 50.90 | 59.10 | 56.20 | 30.90 | 33.60 |
| | AmoebaLLM† (Ours) | **35.90** | **52.36** | 64.70 | 65.70 | 51.80 | **60.80** | **57.10** | 31.00 | **35.40** |

**Models.** We apply our method to LLaMA2 7B [1] and Vicuna 7B v1.5 [41].

**Evaluation.** Following our baselines [7, 9, 8, 6], we leverage lm-evaluation-harness [42] to measure the zero-shot accuracy on 7 commonsense reasoning datasets, including BoolQ [43], PIQA [44], HellaSwag [45], WinoGrande [46], ARC-easy [47], ARC-challenge [47], and OpenbookQA [48]. We also benchmark on the factual knowledge dataset MMLU [49].

**AmoebaLLM's setting.** We set the depth choices to range from 20 to 32 and the width remaining ratios as {1, 7/8, 3/4, 5/8, 1/2} by default unless specifically stated. In each iteration, we sample 4 subnets for joint training and select 2 LoRAs out of a total of 5 LoRAs in SMoL for each subnet.

## 4.2 Benchmark with SOTA LLM Compression Methods

We benchmark our AmoebaLLM against SOTA LLM width/layer pruning methods on LLaMA2 7B/Vicuna 7B v1.5 in Tab. 1/Tab. 2, respectively. Note that the subnets produced by AmoebaLLM are instantly extracted from the same one-for-all fine-tuned LLM, where the (depth, width scale) settings for 80%/65%/50% remaining ratios, determined by the final subnet search in Sec. 3.4, are (30, 0.875)/(28, 0.75)/(22, 0.75), respectively. We also provide the per-subnet fine-tuned counterparts of our delivered subnets, denoted as AmoebaLLM†, to compare one-for-all and individual fine-tuning.

**Benchmark under comparable model sizes.** As shown in Tab. 1 and Tab. 2, we observe that (1) the subnets instantly extracted by AmoebaLLM can achieve higher MMLU accuracy compared to all baselines, suggesting that our method better preserves the factual knowledge acquired during pre-training, as further analyzed in Sec. 4.3 and Sec. 4.4; (2) AmoebaLLM's delivered subnets, extracted from the same model, can also achieve better or comparable average commonsense reasoning accuracy compared to the strongest baselines, each trained separately; (3) AmoebaLLM† achieves the best performance across all metrics and tasks compared to the baselines, indicating that AmoebaLLM† can serve as a new SOTA LLM compression framework in addition to its one-for-all functionality, thus advancing the achievable accuracy-efficiency trade-off; (4) compared to our per-subnet fine-tuned variant AmoebaLLM†, the instantly delivered subnets achieve comparable performance. This demonstrates the effectiveness of our one-for-all fine-tuning scheme, as further ablated in Sec. 4.3.

Table 3: Ablation Study on the effectiveness of the DP-based depth shrinking on LLaMA2 7B.

| Calib. Data | Method | 24 | 23 | 22 | 21 | 20 | 19 | 18 | 17 | 16 |
|---|---|---|---|---|---|---|---|---|---|---|
| Wikitext2 ↓ | Unreasonable [33] | 12.25 | 13.72 | 18.61 | 31.35 | 44.28 | 57.34 | 80.29 | 120.57 | 188.27 |
| | ShortenLLaMA [9] | 11.75 | 13.18 | 17.83 | 29.61 | 39.42 | 53.27 | 71.80 | 106.63 | 154.69 |
| | **Ours** | **11.65** | **12.77** | **17.59** | **20.06** | **23.77** | **28.83** | **38.70** | **70.87** | **95.16** |
| MMLU (%) ↑ | Unreasonable [33] | 40.8 | 39.5 | 37.5 | 33.5 | 34.5 | 34.0 | 30.3 | 30.6 | 25.8 |
| | ShortenLLaMA [9] | 42.0 | 34.7 | 35.1 | 32.1 | 32.0 | 33.5 | 33.7 | 29.3 | 26.6 |
| | **Ours** | **46.2** | **44.8** | **44.6** | **44.1** | **41.2** | **41.3** | **43.1** | **34.7** | **28.9** |

Table 4: Ablation Study on different components in our AmoebaLLM on LLaMA2 7B.

| Method | 32 | | 24 | | 20 | |
|---|---|---|---|---|---|---|
| | Wikitext2 ↓ | MMLU (%) ↑ | Wikitext2 ↓ | MMLU (%) ↑ | Wikitext2 ↓ | MMLU (%) ↑ |
| Per-subnet ft. | **5.54** | 46.4 | **10.57** | 41.9 | **15.94** | 41.7 |
| - SMoL (+full model) | 5.82 | 46.6 | 38.48 | 32.6 | 167.74 | 36.4 |
| - SMoL (+LoRA) | 6.97 | 40.6 | 12.71 | 40.0 | 19.12 | 37.9 |
| - Loss-mag. Balancing | 6.77 | 42.0 | 12.63 | 40.1 | 18.19 | 39.3 |
| Full (AmoebaLLM) | 6.36 | **47.2** | 12.40 | **45.1** | 18.15 | 41.0 |

**Benchmark accuracy-latency trade-offs on real devices.** We further benchmark the achieved trade-off between average common-sense reasoning accuracy and measured latency of LLaMA2 7B using MLC-LLM and PyTorch as the deployment flows on an NVIDIA A5000 GPU, following the settings in Sec. 2. For our method, we select the subnet shape that favors the hardware characteristics based on the profiling in Sec. 2 from the one-for-all fine-tuned LLM. As shown in Fig. 3, we observe that (1) our method consistently achieves the best trade-off on both deployment scenarios; and (2) al-

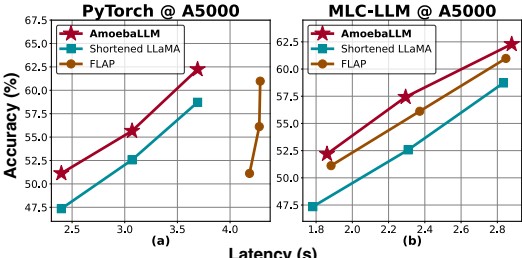

Figure 3: Benchmark AmoebaLLM's achieved accuracy-latency trade-offs with SOTA LLM compression methods on an NVIDIA A5000 GPU.

though FLAP achieves higher accuracy than Shortened LLaMA under comparable model sizes, its real-device speed is limited when using vanilla PyTorch, aligning with observations in [9]. In contrast, our method can instantly deliver subnets that favor the hardware/deployment flow characteristics, thus enjoying both high accuracy and real-device friendliness.

### 4.3 Ablation Study: Effectiveness of Each Component

We perform ablation studies to validate the effectiveness of each component of AmoebaLLM.

**The DP-based depth shrinking strategy.** We benchmark our DP-based strategy against two existing LLM layer pruning methods, Shortened LLaMA [9] and Unreasonable [33], on LLaMA2 7B. Specifically, we employ the three methods to select important layers using Wikitext2/MMLU with PPL/accuracy as calibration metrics, respectively. We directly report the achieved PPL/MMLU accuracy after calibration under various layer remaining ratios *without fine-tuning* to indicate their effectiveness in identifying important layers

*Observations and analysis.* As shown in Tab. 3, we observe that our DP-based strategy outperforms the other two strategies on both calibration datasets and metrics, especially under small remaining ratios, e.g., a +9.4% MMLU accuracy and a -33.1 PPL over the strongest baseline when remaining 18 layers. This demonstrates the superiority of our method over the two baselines in selecting important layers that optimize the target calibration metric, thus significantly contributing to knowledge preservation.

*Remark.* This set of experiments supports our analysis in Sec. 3.2 that the superiority of our method arises from its consideration of different layers' joint contributions, rather than focusing on single-layer importance [9], and its avoidance of reliance on pre-defined rules [33], thus ensuring generality.

**The SMoL adapter.** To assess the efficacy of our SMoL adapter, we substitute it with full model fine-tuning or the standard LoRA [18] and benchmark it against our AmoebaLLM with SMoL as

Table 5: Ablation Study on the selection of calibration datasets on LLaMA2 7B.

| Calib. Data | 32 | | 24 | | 20 | |
|---|---|---|---|---|---|---|
| | Wikitext2 ↓ | MMLU (%) ↑ | Wikitext2 ↓ | MMLU (%) ↑ | Wikitext2 ↓ | MMLU (%) ↑ |
| BookCorpus | 5.64 | 46.4 | 16.52 | 26.8 | 31.70 | 27.0 |
| Wikitext2 | 5.64 | 46.4 | **11.92** | 26.4 | **23.77** | 24.4 |
| MMLU | 5.64 | 46.4 | 293.01 | **45.8** | 1338.68 | **41.2** |
| BookCorpus (ft) | 6.55 | 45.4 | 10.55 | 29.7 | 14.36 | 25.7 |
| Wikitext2 (ft) | 6.41 | **43.4** | **9.58** | 32.6 | **13.26** | 23.6 |
| MMLU (ft) | **6.36** | 47.2 | 12.40 | **45.1** | 18.15 | **41.0** |

well as the per-subnet fine-tuning variant. To more clearly demonstrate its efficacy, this experiment is conducted under a depth-shrinking-only setting, which only enables depth shrinking ranging from 20 to 32 layers during one-for-all fine-tuning. We select three layer configurations, covering both the largest and smallest ones, to report the performance.

*Observations and analysis.* As illustrated in Tab. 4, we observe that (1) full model fine-tuning on a relatively small corpus results in suboptimal performance with large PPL; (2) employing the standard LoRA stabilizes the one-for-all fine-tuning compared to full model fine-tuning, albeit with notable performance reductions in larger subnets, e.g., a 5.8% MMLU accuracy drop in the largest subnet compared to per-subnet fine-tuning; (3) when equipped with our SMoL, the MMLU accuracy of all subnets is substantially enhanced, e.g., a +6.6% improvement on the largest subnet compared to the LoRA case, even exceeding that of the per-subnet counterparts. This demonstrates the essential capability of our AmoebaLLM to deliver high-quality LLM subnets across a wide range of accuracy-efficiency trade-offs.

**The loss-magnitude balancing scheme.** As shown in Tab. 4, we further disable the loss-magnitude balancing scheme and observe a significant MMLU accuracy drop in larger subnets. This supports our analysis in Sec. 3.4 that the larger losses from smaller subnets may dominate the overall objective, thereby impairing the fine-tuning of larger subnets and underscoring the necessity of our method.

## 4.4 Ablation Study: The Selection of Calibration Datasets

We conduct an ablation study on the choice of calibration datasets for our DP-based depth shrinking introduced in Sec. 3.2. Specifically, we use accuracy on the training set of MMLU [49] and PPL on the training sets of Wikitext2 [50]/BookCorpus [51] as target metrics. We report the evaluation metrics, including MMLU test accuracy and Wikitext2 test PPL, under different layer remaining ratios both after calibration and after one-for-all fine-tuning under a depth-shrinking-only setting.

*Observations.* As shown in Tab. 5, we observe that (1) after calibration without fine-tuning, the subnets perform well on the evaluation metric for which they were calibrated and underperform in terms of the other metric; (2) after fine-tuning, the subnets calibrated using PPL continue to perform poorly on MMLU accuracy, indicating a severe loss of factual knowledge. In contrast, the subnets calibrated using MMLU accuracy achieve notably lower PPL compared to before fine-tuning, even on par with the subnets calibrated using PPL, while still maintaining high MMLU accuracy.

*The key insight.* This set of experiments indicates that the loss of factual knowledge during compression is hard to restore during fine-tuning, echoing the observations in [23], while the language modeling capability is easier to recover through fine-tuning. As such, we adopted MMLU as the calibration dataset throughout the previous experiments, and we believe this insight could inspire future LLM compression frameworks and calibration metrics.

## 4.5 Limitations and Future Work

One limitation of our work is that due to the limited fine-tuning data and resources, our methodology is applied to parameter-efficient fine-tuning, which mitigates gradient conflicts under small data conditions while limiting the achievable accuracy-efficiency trade-off. We anticipate that by leveraging more extensive fine-tuning data beyond our current use of Alpaca [40] and extending our design insights regarding subnet selection and gradient conflict mitigation, more aggressive accuracy-efficiency trade-offs can be achieved, which will be the focus of our future work.

# 5  Related Work

**Large language models.** Before the advent of LLMs, transformer-based language models [52, 53, 54, 55] demonstrated their ability to effectively analyze relationships among tokens in complex input sequences, facilitated by the attention mechanism [52]. These models also exhibit notable scalability [56, 57, 58] with respect to model size and the scale of pre-training datasets. This decent scalability has led to the emergence of LLMs, such as GLM [59], OPT [60], BLOOM [61], the Llama family [1, 1, 2], Gemma [3], and GPT-4 [4], which exhibit impressive zero-shot and few-shot in-context learning capabilities. However, these LLMs often feature billions of parameters and prohibitive computation complexity, which limits their widespread use across diverse platforms.

**Large language model compression.** To facilitate efficient deployment of LLMs in real-world applications, existing works primarily focus on compressing LLMs by extending traditional compression techniques such as knowledge distillation [62, 63], quantization [64, 65, 66, 67, 68, 69], system acceleration [70, 71], and pruning [6, 5, 72]. Our work is most closely related to LLM pruning. Along this direction, early works [5, 6] employ unstructured and semi-structured pruning [73] by zeroing out connections among neurons. Despite their plausible performance, these methods require specialized support to achieve real-device speedup. To benefit commodity platforms, structured LLM pruning methods remove more coarse-grained components, such as all connections related to a single neuron [7, 8, 10] or even entire layers [9, 33]. For instance, LLM-Pruner [7] and FLAP [8] reduce LLM width by eliminating identified redundant neurons, while Sheared-LLaMA [10] learns a set of binary masks to reduce both the width and depth of LLMs. However, these methods either focus on a single dimension of compression (i.e., depth or width) with limited efficiency improvements, or they require a costly fine-tuning process for each target configuration and platform. In contrast, our AmoebaLLM can instantly extract subnets of arbitrary shapes that reach the accuracy-efficiency frontier, thus facilitating rapid deployment across devices.

**One-for-all networks.** Slimmable networks [74, 75] are pioneering works that enable a single model to operate at varying widths. Follow-up works [11, 76, 12, 13, 14, 15] further extend this approach to train more general one-for-all networks with switchable depth and width, thus enabling tasks like neural architecture search. In particular, BigNAS [11] builds one-for-all networks using a sandwich sampling strategy that samples a random set of subnets and jointly trains them in each iteration through in-place distillation, where the largest model guides the learning of the smaller ones. This approach has been inherited by subsequent works [12, 13, 14]. Additionally, this idea has been extended to any-precision networks [77, 78, 79] that allow switchable precision at runtime. A very recent work [80] has further extended this concept to any-precision LLMs.

Nevertheless, directly applying these methods to the depth and width of pre-trained LLMs would likely fail because their subnet sampling strategies often select the first layers of a model or the first channels of a layer, which are intended for models trained from scratch. This approach is unsuitable for pre-trained LLMs, as it may omit layers or neurons containing crucial knowledge. Our AmoebaLLM framework addresses these challenges by developing three key components: the subnet selection strategy, the trainable adapter design, and the fine-tuning objective. One concurrent work [81] also aims to train many-in-one LLMs that support instant subnet derivation, targeting a full-model continual training setting on 90 billion tokens. In contrast, our method targets a parameter-efficient tuning setting with only 8.5 million fine-tuning tokens.

# 6  Conclusion

In this work, we present a framework called AmoebaLLM that grants a given LLM the capability to instantly deliver subnets of arbitrary shapes, which achieve the accuracy-efficiency frontier and can be extracted immediately after a one-time fine-tuning. This is achieved by the development of three dedicated components that enable AmoebaLLM's one-for-all fine-tuning scheme: a knowledge-preserving subnet selection strategy, an SMoL adapter, and an in-place distillation objective with loss-magnitude balancing. Extensive experiments validate that our AmoebaLLM framework can deliver efficient LLMs with instantly serviceable subnets of any shape, which outperform SOTA LLM compression techniques in terms of the accuracy-efficiency trade-off. We believe this work is promising to facilitate the wider use of existing and emerging public LLMs by making them instantly deployable on varying platforms and applications, and by providing a new perspective on efficient LLM deployment, thus inspiring future solutions.

## Acknowledgement

The work is supported by an National Science Foundation (NSF) CAREER award (Award number: 2345577) and CoCoSys, one of the seven centers in JUMP 2.0, a Semiconductor Research Corporation (SRC) program sponsored by DARPA.

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
