# OpenReview forum: "AmoebaLLM: Constructing Any-Shape Large Language Models for Efficient and Instant Deployment"
_NeurIPS.cc/2024/Conference — NeurIPS 2024 poster_

### Official Review · Reviewer_U1ac · 2024-07-11

**Soundness:** 3
**Presentation:** 3
**Contribution:** 3
**Rating:** 5
**Confidence:** 3

**Summary:**

The authors proposed a novel pipeline for once-for-all training multiple subnets in a supernet LLM under different resource constraints. The  entire pipeline consists of a knowledge preserving subnet selection utilizing DP to sample depth and width and a new LoRA to resolve the gradient conflicts during training multiple subnets jointly. The experimental results on two widely used decoder-only architectures show the efficacy of the pipeline.

**Strengths:**

- The paper is well-motivated and well-written. To deploy the LLM on multiple resource-constrained devices, fine-tuning multiple subnets jointly is a promising direction to save the computation and training efforts.
- Resolving the complexity when jointly selecting the removal layers in layer-wise pruning is a remaining question for a long time. Although the authors did not entirely solve the problem, the proposed DP solution may potentially also benefit other layer-wise network architectures, e.g., CNN, encoder model. etc.

**Weaknesses:**

- Some technical details are not clear to me. For example, how are the depth and width defined in decoder models? Is a self-attention block a layer? Or does the authors consider FFN and MHSA as two individual layers? In terms of the width, it is clear for FFN but is that the number of heads in MHSA?
- What metric is used as the importance in subnets sampling? I assume that that importance is not derived from gradients, since DP is done layer-by-layer in forward manner. If so, will that metric work better than gradient-based metrics?
- The authors introduce a new LoRA in the paper. Does the authors freeze the backbone LLM and only tune the adapters? If not, how does the authors avoid the co-adaptation of backbone and adapters? Since the original LoRA is proposed to preserve the learned knowledge and fine-tuning only low-rank adaptation for a small amount of new data.
- Also are the adapters only added in MHSA? If so, can the authors provide some theoretical or experimental results to show why FFN layers do not suffer from gradient conflicts?

**Questions:**

See above

**Limitations:**

Yes

---

> ### Author Rebuttal · Authors · 2024-08-07
>
> Thank you for recognizing the value of our work and for your constructive comments! We have addressed all your concerns below.
>
> **1. The definition of depth and width in decoder models**
>
> Thank you for pointing this out! In this work, depth is defined as a whole self-attention block, including both an MHSA and an FFN.
>
> For width, your understanding is correct. Specifically, for FFNs, the width is defined as channels. For MHSA, the width is defined as the number of attention heads, since all channels within one attention head will be kept or pruned as a whole for real-device efficiency, following the definitions and strategies in previous structured LLM pruning works ([6][8] cited in our manuscript).
>
> We will clarify this in the final version.
>
> ---
> **2. The metric used as the importance in subnets sampling of our DP algorithm**
>
> As elaborated in Section 5.4 of our manuscript, we adopt the MMLU accuracy, which serves as an indicator for the encoded factual knowledge, as the DP metric for subnet selection. This is based on the key insight that the loss of factual knowledge during compression is hard to restore during fine-tuning and thus should be prioritized in subnet selection, while the language modeling capability is easier to recover through fine-tuning.
>
> In addition to the benchmark between our DP algorithm and existing layer pruning methods in Table 2 of our manuscript, where our method consistently outperforms all baselines, we follow your suggestion to further benchmark our DP algorithm with gradient-based metrics. Specifically, the gradient-based metric we tried is defined as the average gradients received by all MLPs falling within one layer (defined as a whole self-attention block), where layers with lower average gradients are potentially less important and thus pruned. Following the evaluation setting of Table 2 of our manuscript, we summarize the achieved Wikitext2 PPL and MMLU accuracy when remaining different numbers of layers on LLaMA 2 7B in the table below.
>
> | Calib. Data | # Remained Layers | 24 | 23 | 22 | 21 | 20 | 19 | 18 | 17 | 16 |
> |:---:|:---:|:---:|:---:|:---:|:---:|:---:|:---:|:---:|:---:|:---:|
> | Wikitext2 (PPL) | Gradient-based | 12.689 | 16.394 | 21.813 | 25.209 | 67.262 | 326.666 | 575.863 | 2295.172 | 2140.926 |
> |  | Ours | 11.66 | 12.77 | 17.59 | 20.06 | 23.77 | 28.83 | 38.70 | 70.87 | 95.16 |
> | MMLU (Acc %) | Gradient-based | 24.4 | 26.2 | 24.9 | 24.6 | 23.9 | 234.0 | 24.8 | 24.8 | 24.2 |
> |  | Ours | 46.2 | 44.8 | 44.6 | 44.1 | 41.2 | 41.3 | 43.1 | 34.7 | 28.9 |
>
> We can observe that our method consistently outperforms gradient-based metrics. According to our analysis in Section 4.2, this may be because (1) gradient-based metrics cannot effectively evaluate the joint contributions of different layer combinations like our DP method; and (2) gradient-based metrics cannot effectively reflect the encoded factual knowledge.
>
> ---
> **3. Whether the backbone is frozen and only the adapters are tunable**
>
> Yes, we freeze the backbone and only tune the adapters. This is because we find that one-for-all fine-tuning of the whole backbone on low-resource fine-tuning data results in suboptimal performance of particular subnets, as shown in Table 3 of our manuscript.
>
> ---
> **4. Where the adapters are added**
>
> We follow the settings in QLoRA [1] to add adapters to all linear layers in the pre-trained LLM, including both MHSAs and FFNs.
>
> To study the relative importance of adapters in MHSA and FFNs, we further explore variants where adapters are added only to one of them on top of our method and summarize the results in the table below.
>
> | Remaining Ratio | Attn only |  | FFN only |  | Attn + FFN |  |
> |:---:|:---:|:---:|:---:|:---:|:---:|:---:|
> |  | Wikitext2 | MMLU (%) | Wikitext2 | MMLU (%) | Wikitext2 | MMLU (%) |
> | 80% | 13.31 | 44.6 | 12.14 | 45.6 | 10.34 | 47.4 |
> | 65% | 14.11 | 38.3 | 13.63 | 39.1 | 11.76 | 40.3 |
> | 50% | 19.23 | 32.5 | 16.98 | 33.1 | 15.58 | 34.0 |
>
> We find that (1) adding adapters to both layers achieves the best task performance, and (2) adding adapters to FFNs results in better task performance compared to adding them to MHSAs. This indicates that FFNs may play a more effective role in adjusting the output token distributions after compression.
>
> [1] “QLoRA: Efficient Finetuning of Quantized LLMs”, T. Dettmers et al., NeurIPS’23.

---

> ### Author Response · Authors · 2024-08-11
>
> Dear Reviewer,
>
> We sincerely appreciate the time you dedicated to providing valuable feedback on our paper. In this author response, we have addressed all of your initial concerns. If you have any further questions or concerns, we are happy to discuss them with you. Additionally, we welcome any new suggestions or comments you may have!
>
> Best,
>
> The Authors of Paper #13852

---

### Official Review · Reviewer_wNCb · 2024-07-15

**Soundness:** 3
**Presentation:** 3
**Contribution:** 3
**Rating:** 7
**Confidence:** 3

**Summary:**

This paper proposes AmoebaLLM: a one-for-all fine-tuning and compression framework for delivering pruned and accurate subnets from a pre-trained LLM at various pruning (both depth and width) ratios without the need to fine-tune individual subnets. AmoebaLLM consists of two components: 1) a dynamic programming-based approach to layer selection for selecting subnets of certain depth that takes into account various layer interactions, as opposed to greedy-layer based approaches; 2) after the subnets are defined, a one-for-all fine-tuning strategy is proposed based on gating multiple LoRA heads that are subnet-specific. The training strategy uses a hybrid loss where the largest subnet (always sampled) computes its loss with the ground truth, and all subsequently sampled subnets compute a distilled loss from the largest subnet. The final loss is computed by weighing each subnet's loss appropriately. To demonstrate the efficacy of their method, the authors perform experiments on both LLama 2 7B and Vicuna 7b v1.5 and compare against prior LLM pruning techniques.

**Strengths:**

- The paper tackles a challenging and relevant problem: compressing pre-trained LLMs with various complexities while maintaining their performance
- The paper is well written and easy to follow
- The method overall is novel, and performs better than other pruning methods.
- The paper provides interesting insights into how different LLM metrics are impacted by model compression, e.g., Sec 5.4 shows that MMLU is a better metric for calibration.

**Weaknesses:**

- The claims that full model fine-tuning suffers from severe gradient conflicts appear rather weak. In line (233) the authors state "As analyzed in Sec. 2 and demonstrated in Sec. 5.3 ... ", whereas Section 2 simply conjectures that this is an issue as it uses phrases like" .. would likely fail .." and "as it may omit layers". I would suggest citing the results from Section 5.3 directly to motivate this issue. As for the results themselves in Table 3, the performance of the full model (second row) does not seem consistent. While it performs poorly on Wikitext2, the MMLU performance remains comparable with the SMoL baseline, and seems to improve when pruning more (24 to 20) as opposed to all other methods. It is not clear how these results lead to the conclusion that there is a "gradient conflict". How would the same experiments look when using the Vicuna model? and how would the other evaluation metrics look?
- Using dynamic programming in the context of neural network pruning is not new (https://arxiv.org/pdf/2308.10438, https://ieeexplore.ieee.org/document/9116287 to name a few). The authors should cite and discuss these works, even if they do not tackle LLMs specifically.
- In line 199, the authors claim that the DP complexity is O(MN). This is a bit misleading, as it ignores the complexity of computing P(n,m) which I argue is also dependent on M and N. Furthermore, the actual run-time of the DP step is not discussed, is it too negligible?
- In Table 1, comparing with AmoebaLLM$^\dagger$ is not a fair as it is trained for longer (on top of the training performed in regular AmoebaLLM). How will the other methods compare with the same amount of training?
- It would improve reading clarity if the authors explicitly mention how many layers do the models being studied contain, as simply mentioning how many layers are being pruned in depth shrinkage is hard to gauge without that context. This should also be mentioned in the Tables.

**Questions:**

- AmoebaLLM produces different subnets, some may have the same parameter count due to different width/depth shrinkage ratios, but it is not clear how a model is selected for a certain experiment (e.g. Fig 3). Is it simply enumerating all the subnets and picking the fastest  one? Are the profiling results obtained from Section 3 being used for this, or is that purely for motivating the problem?
- What ranks are used for LoRA?

**Limitations:**

The authors have addressed the limitations of their proposed work.

---

> ### Author Rebuttal · Authors · 2024-08-07
>
> Thank you for recognizing the novelty and interesting insights of our work, as well as for your constructive comments! We have addressed all your concerns below.
>
> **1. Cite the results from Section 5.3 to motivate the issue of full model fine-tuning**
>
> Thank you for the suggestion! We believe what you suggested is a more direct way to motivate the adoption of adapters and will follow your recommendation in the final version.
>
> ---
> **2. Clarify the inconsistent results of Table 3**
>
> Thanks for the interesting question! The reasons for the comparable MMLU performance, which corresponds to the understanding of factual knowledge, between full model fine-tuning and other parameter-efficient settings are two-fold:
>
> *(1)* As highlighted and analyzed in Section 5.4 of our manuscript, as well as echoed by recent observations regarding the role of LLM fine-tuning [1][2][3], new factual knowledge can hardly be gained during fine-tuning, and the loss of factual knowledge during compression is hard to restore during fine-tuning. As such, the MMLU performance is mainly determined by the selection of subnets, which is the same across settings in Table 3 of our manuscript, thus making the MMLU accuracy relatively comparable.
>
> *(2)* Following lm-evaluation-harness ([69] cited in our manuscript), the MMLU accuracy is calculated by picking the largest logit among the four logits corresponding to A/B/C/D, instead of all logits. As such, even if the model performs poorly in language modeling on the MMLU dataset, it still has chances to maintain comparable accuracy. To validate this, we further report the PPL achieved on the MMLU dataset, where the target texts are constructed by question plus correct answer contents. As shown in the table below, which is an extended version from Table 3 of our manuscript, we can observe that: Similar to the PPL on Wikitext2, full model fine-tuning suffers from a notable PPL increase on the MMLU dataset, indicating its failure in language modeling on the MMLU dataset.
>
> | Method | 32 |  |  | 24 |  |  | 20 |  |  |
> |:---:|:---:|:---:|:---:|:---:|:---:|:---:|:---:|:---:|:---:|
> |  | Wikitext2 (PPL) | MMLU (PPL) | MMLU (Acc %) | Wikitext2 (PPL) | MMLU (PPL) | MMLU (Acc %) | Wikitext2 (PPL) | MMLU (PPL) | MMLU (Acc %) |
> | Per-subnet ft. | 5.54 | 20.62 | 46.4 | 10.57 | 45.59 | 41.9 | 15.94 | 65.9 | 41.7 |
> | - SMoL (+full model) | 5.82 | 27.02 | 46.6 | 38.48 | 291.11 | 32.6 | 167.74 | 1349.17 | 36.5 |
> | - SMoL (+LoRA) | 6.97 | 32.75 | 40.6 | 12.71 | 81.9 | 40.0 | 19.12 | 122.67 | 37.9 |
> | - Loss-mag. Balancing | 6.77 | 26.34 | 42.0 | 12.63 | 59.81 | 40.1 | 18.19 | 91.07 | 39.3 |
> | **Full** | **6.36** | **25.02** | **47.2** | **12.40** | **55.8** | **45.1** | **18.15** | **84.27** | **41.0** |
>
> Following your suggestion, we have also provided the benchmark with the full model fine-tuning variant of our method on Vicuna 7B below. The results are consistent with previous observations: (1) full model fine-tuning leads to poor language modeling capabilities across both datasets, and (2) the MMLU accuracy of full model fine-tuning does not suffer a very drastic drop but is notably worse than our method.
>
> |  | 32 |  |  | 24 |  |  | 20 |  |  |
> |:---:|:---:|:---:|:---:|:---:|:---:|:---:|:---:|:---:|:---:|
> | Method | Wikitext2 (PPL) | MMLU (PPL) | MMLU (Acc %) | Wikitext2 (PPL) | MMLU (PPL) | MMLU (Acc %) | Wikitext2 (PPL) | MMLU (PPL) | MMLU (Acc %) |
> | Full Model Finetuning | 7.29 | 25.52 | 46.9 | 40.7 | 164.67 | 39.3 | 134.07 | 882.82 | 37.1 |
> | Ours | 6.85 | 23.27 | 47.9 | 11.87 | 49.51 | 46.1 | 14.77 | 67.73 | 45.4 |
>
>
> [1] “Does fine-tuning llms on new knowledge encourage hallucinations?”, Z.  Gekhman et al., arXiv’24.
>
> [2] “LIMA: Less Is More for Alignment”, C. Zhou et al., NeurIPS’23.
>
> [3] “R-Tuning: Instructing Large Language Models to Say ‘I Don’t Know’”, H. Zhang et al., NAACL’24.
>
> ---
> **3. Missing citations**
>
> Thank you for providing these related works! We will cite and comment on them in the final version.
>
> ---
> **4. The DP complexity and run-time**
>
> Thank you for pointing this out! You are right that the DP complexity should take the varying number of layers involved in evaluating *P(n, m)* into consideration. The updated DP complexity is *O(MN(M-N))*.
>
> For the run-time of our DP algorithm, when applying it to LLaMA2 7B on an NVIDIA A5000 GPU in our case, where N=32 and M=16, the total consumed time is 1 hour.
>
> We will add the above information in our final version.

---

> ### Author Response · Authors · 2024-08-07
> **Author Response - Part 2**
>
> **5. Benchmark with baselines with the same amount of training time**
>
> We first clarify that in Table 1, we follow a standard criterion in the literature, i.e., ensuring the same number of fine-tuning tokens across both the baselines and our method.
>
> Following your suggestion, we further train the baselines with the same amount of training time as our method, which corresponds to 20k training iterations on Alpaca-gpt4. As shown in the table below, we report the MMLU accuracy as well as the average accuracy across 7 tasks, following the task list of Table 1. We observe that (1) the baselines fine-tuned with 20k iterations generally maintain comparable performance with those trained with 10k iterations, potentially because 10k iterations is sufficient for fine-tuning on Alpaca-gpt4; and (2) our method still outperforms all baselines across various tasks.
>
> | **Remaining  Ratio** | **Method** | **Training Iterations** | **Training Time** | **MMLU (%)** | **Average Acc (%)** |
> |:---:|:---:|:---:|:---:|:---:|:---:|
> | 80% | FLAP | 10k | 15h | 40.21 | 60.98 |
> |  |  | 20k | 30h | 38.62 | 61.40 |
> |  | Shortened LLaMA | 10k | 15h | 26.45 | 58.72 |
> |  |  | 20k | 30h | 26.55 | 59.63 |
> |  | Ours  | 10k | 30h | **40.70** | **62.29** |
> | 65% | FLAP | 10k | 15h | 33.28 | 56.12 |
> |  |  | 20k | 30h | 35.4 | 55.93 |
> |  | Shortened LLaMA | 10k | 15h | 24.89 | 52.57 |
> |  |  | 20k | 30h | 24.70 | 53.95 |
> |  | Ours  | 10k | 30h | **36.00** | **56.96** |
> | 50% | FLAP | 10k | 15h | 27.67 | 51.12 |
> |  |  | 20k | 30h | 27.55 | 51.32 |
> |  | Shortened LLaMA | 10k | 15h | 24.76 | 47.35 |
> |  |  | 20k | 30h | 25.10 | 49.22 |
> |  | Ours  | 10k | 30h | **30.60** | **52.19** |
>
>
> **6. The number of remained depth and width in Table 1**
>
> Thank you for the suggestion! The (depth, width scale) for AmoebaLLM with 80%/65%/50% remaining ratios in Table 1 are (30, 0.875)/(28, 0.75)/(22, 0.75), respectively. We will follow your suggestion to add this information to Table 1 to improve readability.
>
> ---
> **7. The detailed subnet selection strategy**
>
> The profiling results in Section 3 are purely for motivating the problem. Here is our current subnet selection strategy: we adopt a hierarchical search strategy to deliver subnets from our design space that satisfy the target efficiency constraint, e.g., 50% weight remaining ratios in Table 1 of our manuscript, while maximizing the achievable accuracy. Specifically, we first perform a coarse grid search across uniformly spaced depth and width settings based on a small calibration set, e.g., 20 samples from the MMLU dataset, to identify the subnets that satisfy the given efficiency constraint with maximized accuracy. Next, we perform a more fine-grained grid search within depth/width ranges surrounding the optimal subnet identified in the coarse grid search stage. This process typically evaluates 40 promising subnets and takes no more than 10 minutes on an NVIDIA A5000 GPU.
>
> We empirically find that the above strategy works well at the scale of our target problem and can already deliver high-quality subnets that outperform previous compression methods. We also note that more complex subnet selection strategies, such as the evolutionary search adopted by [11]-[15] cited in our manuscript, can also be employed, which will be our future work.
>
> We will clarify this in the final version.
>
> ---
> **8. The rank of LoRA**
>
> We follow QLoRA [1] and adopt a rank of 64. We will add this information to the final version.
>
> [1] “QLoRA: Efficient Finetuning of Quantized LLMs”, T. Dettmers et al., NeurIPS’23.

---

> > ### Comment · Reviewer_wNCb · 2024-08-13
> >
> > Thank you for the detailed response. The authors have addressed most of my concerns/questions. The additional data/clarifications provided in this rebuttal should be included in the final manuscript as the authors mentioned. I will increase my score (6->7).

---

> > > ### Author Response · Authors · 2024-08-13
> > >
> > > Thank you for taking the time to review our rebuttal responses and for providing positive feedback! We are encouraged to hear that our rebuttal has addressed most of your concerns. Following your suggestion, we will include the additional data and clarifications from this rebuttal in our final manuscript.

---

> ### Author Response · Authors · 2024-08-11
>
> Dear Reviewer,
>
> We sincerely appreciate the time you dedicated to providing valuable feedback on our paper. In this author response, we have addressed all of your initial concerns. If you have any further questions or concerns, we are happy to discuss them with you. Additionally, we welcome any new suggestions or comments you may have!
>
> Best,
>
> The Authors of Paper #13852

---

### Official Review · Reviewer_LhJH · 2024-07-21

**Soundness:** 2
**Presentation:** 4
**Contribution:** 3
**Rating:** 5
**Confidence:** 4

**Summary:**

The paper proposed a new framework, named AmoebaLLM, that adapts any LLMs to achieve optimal efficiency across different platforms and applications. In specific, the framework contains two stages. The first stage denoted as a knowledge-preserving stage, creates a subnet of the LLM by dynamic programming given an arbitrary depth or width. This second stage denoted as the one-for-all finetune stage is to fine-tune the obtained subnet to achieve the optimal performance on the given application (dataset). To finetune the subnet, the paper proposed shape-aware LoRAs to adapt the knowledge and a loss-magnitude balancing to ensure efficient distillation in the second stage. All combined, the method achieves SOTA performance among all the pruning methods and obtains the best accuracy in the same latency in different platforms.

**Strengths:**

1. The target of the paper is practical and interesting. It is indeed a good open problem to implement different LLMs within different constraints of devices and platforms while achieving the optimal trade-off between efficiency and accuracy.

2. This method, the DP-based depth shrinking strategy, could construct an any-shape LLM with reasonable performance, which is promising in different applications.

3. The presentation and writing are good and easy to follow.

**Weaknesses:**

1. About latency in section 5.2:The work claimed that it achieve the accuracy-efficiency frontier. In terms of accuracy, it is convincing as shown in Table 1. However, for efficiency, it seems only Fig.3 is related to this topic yet only shows a trade-off between latency and accuracy. It is not sufficient to evaluate the efficiency purely based on latency.

2. In my understanding, efficiency should also consider the resources consumed in the pruning method. First, the DP algorithm itself has a high computational complexity. Second, the proposed method is somewhat limited to its one-for-all finetune, which is a training process related to a specific dataset, and a manual selection of the subnet, the method is not perfectly efficient. In this case, the author may consider adjusting their claim in efficiency.

3. The selection of the subnet seems to be related to the profiling of the generation latency. It is purely empirical and may be error-prone when it is generalized to other platforms.

**Questions:**

1. What is the definition of the latency?

2. Is it possible to provide the details of selecting the subnet shape that favors the hardware characteristics?

3. Which LLM is used across all the experiments?

**Limitations:**

No negative societal impact is found.

---

> ### Author Rebuttal · Authors · 2024-08-07
>
> Thank you for recognizing our work as practical and interesting, as well as for your constructive comments! We have addressed all your concerns below.
>
> **1. The definition of latency and more real-device efficiency measurement**
>
> We clarify that the latency used in Fig. 3 of our manuscript is start-up latency, i.e., the end-to-end time used to finish the generation of one sample (batch size = 1, sequence length = 128).
>
> Following your suggestion, to provide a more comprehensive evaluation of inference efficiency, we further measure the start-up latency (batch size = 1) and throughput (batch size = 16) for generating/prefilling a sequence with a length of 128, respectively, using the MLC-LLM framework on an NVIDIA A5000 GPU. We provide the achieved accuracy (averaged over the 7 tasks from Table 1 of our manuscript) and efficiency metrics in the table below. We can observe that our method still consistently achieves the best accuracy-efficiency trade-off.
>
> |  |  | Generation |  | Prefilling |  |  |
> |---|---|:---:|:---:|:---:|:---:|---|
> | Remaining Ratio | Method | Start-up latency/s (bs=1) | Throughput (bs=16) | Start-up latency/s (bs=1) | Throughput (bs=16) | Accuracy (%) |
> | 80% | Shortened LLaMA | 2.834 | 5.528 | 0.0405 | 48.810 | 58.72 |
> |  | FLAP | 2.847 | 5.427 | 0.0418 | 45.701 | 60.98 |
> |  | **Ours** | **2.879** | **5.286** | **0.042** | **46.256** | **62.29** |
> | 65% | Shortened LLaMA | 2.308 | 6.785 | 0.0331 | 60.060 | 52.57 |
> |  | FLAP | 2.372 | 6.376 | 0.0349 | 56.437 | 56.12 |
> |  | **Ours** | **2.294** | **6.663** | **0.035** | **56.398** | **57.43** |
> | 50% | Shortened LLaMA | 1.782 | 8.887 | 0.0259 | 77.295 | 47.35 |
> |  | FLAP | 1.883 | 8.056 | 0.0275 | 71.942 | 51.12 |
> |  | **Ours** | **1.860** | **8.124** | **0.0281** | **73.260** | **52.19** |
>
>
> **2. The complexity of our DP algorithm**
>
> To address your concern, we provide the consumed time of our DP algorithm here. As elaborated in Section 4.2 of our manuscript, when targeting the removal of M layers out of all N layers, the total number of evaluations required by the DP algorithm is M(N-1)/2. For each evaluation, as mentioned in Section 5.4, we measure the MMLU accuracy of 20 samples, which takes about 15 seconds on average on an NVIDIA A5000 GPU for LLaMA2 7B.
>
> As such, when applying our DP algorithm to LLaMA2 7B in our case, where N=32 and M=16, the total consumed time is about 1 hour, which is about 1/15 of the LoRA fine-tuning time of our baselines in Table 1. It is worth noting that constructing the DP table is a one-time effort for each pre-trained LLM model and does not need to be repeated for each target subnet configuration.
>
> ---
> **3. The overall training efficiency of one-for-all fine-tuning**
>
> First, we clarify that our one-for-all fine-tuning is not dataset-specific; instead, we perform a generic fine-tuning on alpaca-gpt4, as mentioned in Section 5.1 of our manuscript, and then evaluate the subnets with varying shapes derived from the fine-tuned model across different test datasets and tasks, as shown in Table 1 of our manuscript. In addition, the selection of subnets can be automatically performed within 10 minutes, as elaborated in our response to Question 5.
>
> Furthermore, beyond the inference efficiency analyzed in Section 5.2 of our manuscript as well as in Question 1, the advantage of our method in training efficiency is that it requires a constant *O(1)* training time with respect to the number (*N*) of subnets with varying efficiency constraints to be delivered, thanks to the one-for-all fine-tuning. In contrast, previous compression methods require *O(N)* fine-tuning time.
>
> For example, for delivering 10 subnets with varying weight remaining ratios in Table 1 of our manuscript, when fine-tuned on the same number of tokens, all baseline methods require about 150 GPU hours in total on an NVIDIA A5000 GPU, while our method requires only 30 GPU hours.
>
> We will follow your suggestion to emphasize our achieved trade-off between accuracy and inference efficiency, while also clarifying and highlighting our claim regarding our method’s advantage in training efficiency from the above perspective.
>
> ---
> **4. The subnet selection may be error-prone when generalized to various platforms**
>
> We agree that determining the subnet shape is a general problem for hardware-aware compression methods, which can be error-prone and not generalizable across platforms. This issue originates from the fact that theoretical FLOPs cannot always reflect real-device efficiency.
>
> However, our method mitigates this issue by allowing for rapid subnet determination for each target platform, thanks to its delivered one-for-all fine-tuned LLM. Specifically, for each target platform, users can select efficient subnets based on platform-specific profiling and directly obtain the accuracy of these selected subnets from the one-for-all fine-tuned LLM, without the need for per-subnet fine-tuning. This improved training efficiency enables users to quickly identify subnets that hit the Pareto frontier of accuracy-efficiency trade-offs, thus obtaining high-quality subnets efficiently.

---

> ### Author Response · Authors · 2024-08-07
> **Author Response - Part 2**
>
> **5. Detailed subnet selection strategy**
>
> Thank you for pointing this out! Currently, we adopt a hierarchical search strategy to deliver subnets from our design space that satisfy the target efficiency constraint, e.g., 50% weight remaining ratios in Table 1 of our manuscript, while maximizing the achievable accuracy. Specifically, we first perform a coarse grid search across uniformly spaced depth and width settings based on a small calibration set, e.g., 20 samples from the MMLU dataset, to identify the subnets that satisfy the given efficiency constraint with maximized accuracy. Next, we perform a more fine-grained grid search within depth/width ranges surrounding the optimal subnet identified in the coarse grid search stage. This process typically evaluates 40 promising subnets and takes no more than 10 minutes on an NVIDIA A5000 GPU.
>
> We empirically find that the above strategy works well at the scale of our target problem and can already deliver high-quality subnets that outperform previous compression methods. We also note that more complex subnet selection strategies, such as the evolutionary search adopted by [11]-[15] cited in our manuscript, can also be employed, which will be our future work.
>
> We will follow your suggestion and our promise in the abstract to open source all source code and the delivered subnets upon acceptance.
>
> ---
> **6. Which LLMs are used**
>
> As mentioned in Section 5.1 of our manuscript, we employ LLaMA 2 7B and Vicuna 7B v1.5.

---

> ### Author Response · Authors · 2024-08-11
>
> Dear Reviewer,
>
> We sincerely appreciate the time you dedicated to providing valuable feedback on our paper. In this author response, we have addressed all of your initial concerns. If you have any further questions or concerns, we are happy to discuss them with you. Additionally, we welcome any new suggestions or comments you may have!
>
> Best,
>
> The Authors of Paper #13852

---

> > ### Comment · Reviewer_LhJH · 2024-08-11
> >
> > Thank you for your detailed response. My concerns have largely been addressed, and after considering the feedback from other reviewers, I am inclined to support the acceptance of this paper. I have a few additional suggestions:
> >
> > 1. The table provided in the first response should be included in the main paper. It would strengthen the argument regarding the “Frontier of accuracy and efficiency,”, as particularly the method demonstrates state-of-the-art performance across various efficiency metrics.
> > 2. It would be beneficial to include a discussion of the complexity of the DP algorithm and its corresponding running time in the main paper. This addition could alleviate any concerns readers might have about the

---

> > > ### Author Response · Authors · 2024-08-11
> > >
> > > Thank you for taking the time to review our rebuttal responses and for providing positive feedback! We are encouraged to hear that our rebuttal has addressed most of your concerns.
> > >
> > > Following your suggestion, we will include (1) the real-device efficiency table and (2) the complexity and runtime of our DP algorithm, both in our first response, in the final version of our manuscript to strengthen the coherence of our work.

---

### Official Review · Reviewer_TKvU · 2024-07-21

**Soundness:** 3
**Presentation:** 3
**Contribution:** 3
**Rating:** 5
**Confidence:** 5

**Summary:**

To address the problems of diverse resource constraints and deployment flows while using LLM for multiple real-world applications, this paper proposes an AmoebaLLM, featuring a knowledge-preserving subnet selection strategy, a shape-aware mixture of LoRAs and a distillation scheme with loss-magnitude balancing.

**Strengths:**

1. The paper is well-organized.
2. The method achieves SOTA performance on most metrics.
3. The experimental setup is easy to follow.

**Weaknesses:**

Overall, the motivation and ideas of this paper are commendable. However, I have a few concerns:
1. DP-based depth shrinking aims to select suitable layers that can achieve the optimal target metric. Does this mean that the proposed method needs to evaluate the performance for each selection strategy? If so, how long is required for each evaluation? What is the relationship between the number of layers and the overall search time?
2. Another concern is about the optimal subnet. I wonder if the optimal subnet identified in the first phase is truly “optimal.” I believe that the selected subnet is the optimal choice for the initial parameters. However, the proposed method consists of two stages. After the fine-tuning stage, some subnets that were not optimal in the first phase might achieve the best target metrics. I understand that searching subnets with fine-tuning is very expensive, but have the authors considered this issue? Is this correct?
3. I am particularly interested in the training efficiency comparisons, as the authors mentioned that existing methods are not efficient. However, the experimental section does not show such comparisons. Could the authors provide relevant comparisons in Table 1 and Table 2?

**Questions:**

Please see the weaknesses part.

**Limitations:**

Yes.

---

> ### Author Rebuttal · Authors · 2024-08-07
>
> Thank you for recognizing the idea and performance of our work, as well as for your constructive comments! We have addressed all your concerns below.
>
> **1. Whether our DP-based depth shrinking strategy needs to evaluate the performance for each selection strategy and its overhead**
>
> Yes, you are right. Our DP-based strategy needs to evaluate the performance for each selection strategy, as described in Eq. (2) of our manuscript. Specifically, as mentioned in Section 5.4 of our manuscript, we measure the MMLU accuracy of 20 samples for each evaluation, which takes about 15 seconds on average on an NVIDIA A5000 GPU for LLaMA2 7B.
>
> The overall search time is proportional to the total number of layers. Specifically, when targeting the removal of M layers out of all N layers, the total number of evaluations is M(N-1)/2. For LLaMA2 7B in our case, where N=32 and M=16, the overall search time is about 1 hour. It is worth noting that constructing the DP table is a one-time effort for each pre-trained LLM model and does not need to be repeated for each target subnet configuration.
>
> We will add this analysis to the final version.
>
> ---
> **2. Whether the optimal subnets change before and after fine-tuning and the potential of searching subnets with fine-tuning**
>
> Thanks for the insightful question! First of all, we agree that under different subnet selection criteria, the selection of optimal subnets may vary before and after fine-tuning, making smarter strategies like iterative subnet selection and one-for-all fine-tuning promising.
>
> Second, under our subnet selection criteria, i.e., the encoded factual knowledge measured by MMLU accuracy as elaborated in Section 5.4 of our manuscript, we observe a high consistency between the optimal subnets before and after fine-tuning. Specifically, this is because, according to observations in recent works [1][2][3], new factual knowledge can hardly be gained during finetuning, and the loss of factual knowledge during compression is hard to restore during finetuning. As such, the selection of optimal subnets stays consistent when using MMLU accuracy as a subnet selection indicator. This motivates us to adopt a simple-yet-effective two-stage strategy in our AmoebaLLM framework.
>
> [1] “Does fine-tuning llms on new knowledge encourage hallucinations?”, Z.  Gekhman et al., arXiv’24.
>
> [2] “LIMA: Less Is More for Alignment”, C. Zhou et al., NeurIPS’23.
>
> [3] “R-Tuning: Instructing Large Language Models to Say ‘I Don’t Know’”, H. Zhang et al., NAACL’24.
>
> ---
> **3. Training efficiency analysis**
>
> Thanks for constructive comments and questions! Generally, the advantage of our method in training efficiency is that it requires a constant *O(1)* training time with respect to the number (*N*) of subnets with varying efficiency constraints to be delivered, thanks to the one-for-all finetuning. In contrast, previous compression methods require *O(N)* fine-tuning time.
>
> For example, for delivering 3 subnets with varying weight remaining ratios in Table 1 of our manuscript, when fine-tuned on the same number of tokens, all baseline methods require about 45 GPU hours in total on an NVIDIA A5000 GPU, while our method requires 30 GPU hours. Similarly, when delivering 10 subnets, all baseline methods require about 150 GPU hours, while our method still requires only 30 GPU hours, thanks to its constant training complexity.
>
> We will add the above analysis to the final version.

---

> ### Author Response · Authors · 2024-08-11
>
> Dear Reviewer,
>
> We sincerely appreciate the time you dedicated to providing valuable feedback on our paper. In this author response, we have addressed all of your initial concerns. If you have any further questions or concerns, we are happy to discuss them with you. Additionally, we welcome any new suggestions or comments you may have!
>
> Best,
>
> The Authors of Paper #13852

---

> > ### Comment · Reviewer_TKvU · 2024-08-13
> >
> > Thank you for your reply. I will maintain my score.

---

> > > ### Author Response · Authors · 2024-08-13
> > >
> > > Thank you for taking the time to review our rebuttal responses and provide feedback! Any further suggestions or comments you have are welcome!

---

### Decision · Program_Chairs · 2024-09-25

**Decision:**

Accept (poster)

**Comment:**

After the rebuttal, the ratings were: borderline accept, borderline accept, accept, and borderline accept. While none of the reviewers rejected the paper, only one expressed strong support. However, the reviewers did acknowledge the practicality of the problem and the novelty of the proposed method.